# Study on Optimization Strategy for the Composition Transition Gradient in SS 316L/Inconel 625 Functionally Graded Materials

**DOI:** 10.3390/ma17122910

**Published:** 2024-06-14

**Authors:** Qiang Zhu, Xiaoyan Yu, Ping Yao, Youshu Yue, Guopo Kang

**Affiliations:** 1School of Robotics, Guangdong Open University, Guangzhou 510091, China; ysle@gdpi.edu.cn (Y.Y.); gpkang@gdpi.edu.cn (G.K.); 2School of Mechanical and Automotive Engineering, South China University of Technology, Guangzhou 510640, China; 202010100391@mail.scut.edu.cn; 3School of Electrical and Mechanical, Guangdong Polytechnic Normal University, Guangzhou 510635, China

**Keywords:** functionally graded materials, dual-wire arc additive manufacturing, composition transition gradient, optimization strategy, microstructure, mechanical properties

## Abstract

Wire arc additive manufacturing (WAAM) technology enables the fabrication of functionally graded materials (FGMs) by adjusting the wire feed speed of different welding wires in a layer-by-layer manner. This study aimed to produce SS 316L/Inconel 625 FGMs with varying transition compositions using dual-wire arc additive manufacturing (D-WAAM). An optimization strategy for transition gradients was implemented to exclude component regions that are prone to defect formation (notably cracking), as well as to retain other component regions, thereby enhancing the overall mechanical properties of FGMs. The study revealed grain boundary cracking and demonstrated the lowest microhardness and tensile properties within a 20 wt.% Inconel 625 transition gradient zone, which negatively impacts the overall mechanical properties of FGMs. Then, as the content of Inconel 625 in the first transition region increased, cracks disappeared, microhardness increased and better tensile properties were obtained. The most optimal mechanical properties were enriched at 50 wt.% Inconel 625 content. In conclusion, the compositional gradient optimization strategy proves efficacious in eliminating component regions with poor mechanical properties and microdefects, ensuring excellent overall mechanical characteristics of FGMs.

## 1. Introduction

Functionally graded materials (FGMs) are advanced composite materials in which a gradient change in chemical composition or microstructure from one side to the other leads to corresponding changes in these materials’ mechanical properties [1,2]. Additive manufacturing (AM) technology, renowned for its layer-by-layer fabrication process, stands out due to its adaptability, high efficiency and ability to construct intricate components, making it the preferred method for producing metal FGMs [3]. The combination of nickel-based superalloys with stainless steels has emerged as a focal point in the field of FGMs through AM technology. Nickel-based superalloys are appealing for their high strength, good creep properties and long-term durability, as well as their strong resistance to oxidation corrosion, making them suitable for producing vital components such as steam turbines and aeroengines. In contrast, stainless steels are recognized for their outstanding wear resistance, high toughness and low-cost, thus being the most suitable component for supporting and structural materials [4,5,6,7]. For instance, using nickel-based superalloy and stainless steel FGMs, new turbine blades can be manufactured, achieving performance gradients in the root, middle and tip regions of the blades. These blades demonstrate excellent performance in high-temperature, high-stress and corrosive environments [8,9,10]. Currently, several nickel-based superalloy and stainless steel FGMs have been successfully developed, including SS 316L/Inconel 718 [5,11,12], SS 316L/Inconel 625 [9,13], SS 304L/Inconel 625 [10] and SS 321/Inconel 625 [14].

It has been previously established that nickel-based alloy/stainless steel FGMs can be prepared by continuously changing the composition ratio of the two constituents. The emergence of secondary phases, notably Laves and MC phases, affects the mechanical properties of the transition zone, especially within the nickel-based superalloy composition range of 10–50 wt.%. Su et al. [11] prepared SS 316L/Inconel 718 FGMs with different composition gradients (5 wt.%, 10 wt.% and 20 wt.%), using laser metal deposition (LMD) technology. The results revealed the weakest mechanical properties at the composition gradient of 10 wt.%. Concurrently, Li et al. [15] utilized dual-wire arc additive manufacturing (D-WAAM) to create Inconel 625/SS 308L FGMs, and observed cracking at approximately 20 wt.% Inconel 625 content. Fracture morphology and crack location analyses confirmed the occurrence of hot cracking phenomena in the material. Thus, certain compositional ranges are apparently predisposed to a reduction in mechanical performance of the final product. Therefore, enhancing the mechanical properties within these specific compositional regions is crucial for improving the overall mechanical properties of FGMs. In a strategic approach, Kim et al. [5] generated defect-free FGMs transitioning from SS 316L to Inconel 718 through the LMD process, intentionally omitting certain compositional ranges (20–30 wt.%) via optimization of the compositional gradient path. This indicates that the deliberate exclusion of regions with worse mechanical properties can successfully elevate the overall mechanical characteristics of FGMs.

Wire arc additive manufacturing (WAAM) is economically advantageous over powder-based AM processes, regarding lower raw material costs, near-perfect utilization of raw material rates and accelerated manufacturing times, allowing for the production of large and medium-sized parts. Additionally, the WAAM process allows controlled feeding of the wire into the molten pool with minimal spattering and process stability [16]. Nonetheless, the large heat input and the complex thermal history within the WAAM process make it challenging to control the precision of the composition during the fabrication of FGMs compared to the LMD process [17,18,19,20]. In addition, in the WAAM process, the range of compositions with weak mechanical properties may shift due to the influence of remelting during the WAAM process. Additionally, the dimensions of the specimen change during the deposition process and cannot be precisely controlled. Defects may be caused by interlayer oxidation, thermal accumulation or other factors. Therefore, the effectiveness of optimization measures in the WAAM process by eliminating only the poorly performing regions of the FGM has to be further verified.

In this study, SS 316L/Inconel 625 FGMs with various compositional gradients were produced via the D-WAAM technique. Special attention was paid to the ability to eliminate defective compositional zones prone to hot cracking, while preserving the remaining compositional regions. The influence of omitting compositional zones with adverse effects on the mechanical properties throughout FGMs was also investigated.

## 2. Materials and Methods

In this study, SS 304L was utilized as the substrate with dimensions of 100 mm × 200 mm × 10 mm. Before conducting the experiments, the surface of the substrate was mechanically polished to eliminate the oxide film using a grinding machine. The filler wires, SS 316L and Inconel 625, each had a diameter of 1.2 mm. The chemical compositions of the substrate and wires are detailed in Table 1.

Table 2 presents the experimental parameters of the D-WAAM process. The welding current was 110 A and the travel speed ranged from 60 to 65 cm/min. The welding torch height was 15 mm. After completing each layer of deposition, the material was cooled for 30 s before continuing with the next layer. Argon gas (99.99% purity) was used as the shield gas at a constant rate of 15L/min. The optimization strategy for the composition transition gradient is illustrated in Figure 1b. According to the research by Kim and Li et al., defects mainly occur in the 20–30 wt.% Inconel 625 region, and the mechanical properties of the alloy are weak when the Inconel 625 content is in the range of 10–50 wt.% [5,21]. Therefore, to obtain a defect-free FGM with gradient performance, three groups of specimens were designed. These included specimens with compositional gradients of 20 wt.%, 30 wt.% and 50 wt.% Inconel 625. Specimen #1 featured a uniform transition from 100 wt.% SS 316L to 80 wt.% SS 316L/20 wt.% Inconel 625. Specimens #2 and #3, on the other hand, incorporated non-uniform gradients, transitioning from 100 wt.% SS 316L to 70 wt.% SS 316L/30 wt.% Inconel 625 and from 100 wt.% SS 316L to 50 wt.% SS 316L/50 wt.% Inconel 625, respectively. The effectiveness of the optimization strategy was analyzed by comparing the microstructure, composition distribution and mechanical properties of these three groups of specimens. Additionally, the primary causes of performance degradation in the first transition zone with the addition of a small amount of Inconel 625 were described.

A D-WAAM system was chosen as the experimental platform, comprising two power supplies (LORCH S5-Robot MIG), two automatic wire feeders, a FANUC welding robot (M-10iA), a robot control cabinet and two shielding gas systems (Ar, 99.99% purity) (see Figure 1a). A gradient transition from SS 316L to Inconel 625 was realized by modulating the wire feed speeds of both welding wires. It was observed that the compositional gradient was correlated with the wire feed rate and density of the welding wire. Consequently, the design for the compositional gradient of the SS 316L/Inconel 625 FGMs can be summarized in the expression below:(1)$$Wt_{SS\;316L}=\frac{v_{1}dt\cdot ({d_{1}}^{2}\pi /4)\cdot \rho_{1}}{v_{1}dt\cdot ({d_{1}}^{2}\pi /4)\cdot \rho_{1}+v_{2}dt\cdot ({d_{2}}^{2}\pi /4)\cdot \rho_{2}}\;0\leq v_{1}\leq 700$$

The densities of SS 316L and Inconel 625 welding wires are denoted by $\rho_{1}$ and $\rho_{2}$, respectively, which are 7.98 g/cm^3^ and 8.44 g/cm^3^, respectively. The diameters of the SS 316L and Inconel 625 welding wires are represented by $d_{1}$ and $d_{2}$, respectively ($d_{1}$ = $d_{2}\;$ = 1.2 mm); $v_{1}$ and $v_{2}$ are the wire feed speeds of the SS 316L and Inconel 625 welding wires, respectively. Therefore, Equation (1) can be simplified as follows:(2)$$Wt_{SS\;316L}=\frac{7.98v_{1}}{7.98v_{1}+8.44v_{2}}\;0\leq v_{1}\leq 700$$

After completing the experiment, wire electrical discharge machining (EDM) was employed to cut analytical specimens from the fabricated specimens for subsequent analysis, as depicted in Figure 1c. Then, these specimens were cleaned with acetone to remove any surface contaminants, such as oils. Initially, sandpapers with progressively finer grades (180#, 400#, 800# and 1000#) were used for treatment. Subsequently, a more meticulous grinding process was implemented using sandpapers with grit sizes of 2000#, 3000#, 5000# and 7000#. Then, the specimens were polished with a polishing agent with a particle size of 1.0 μm and finally corroded with aqua Regis (HCl:HNO_3_ = 3:1), washed with alcohol and dried for subsequent microstructure analysis. The tensile fracture and microstructure were characterized using a scanning electron microscope (SEM, GEMINISEM 300). The actual composition in the bi-metallic interface of 100 wt.% SS 316L and the first transition region were detected via energy dispersive spectroscopy (EDS). The microhardness distributions along the building direction were obtained by means of a Shimadzu HMV-2T hardness tester under a load of 200 g with a dwell time of 15 s. The test started from the substrate and measured three times at a distance of 2 mm at the same horizontal height. The average value was taken as the microhardness value of the component region. The tensile properties were measured at room temperature using a universal tensile testing machine (CMT5105) at a tensile rate of 1 mm/min. The three tensile specimens labeled H1–H3 were extracted from the left, middle and right sections of the sample, respectively. The average of the three samples selected from different parts of the deposition can represent the tensile result of the entire FGM.

## 3. Results

### 3.1. Microstructure

Figure 2, Figure 3 and Figure 4 illustrate the microstructures of specimens #1, #2 and #3 across various compositional regions. Although the content of Inconel 625 in the first transition zones varied, it exhibited strong bonding with the 100 wt.% SS 316L layer. Additionally, the augmentation of Laves phase precipitates was proportional to the increase in Inconel 625 content, with their morphology transitioning from small particles to blocks.

Specimen #1 (Figure 2) exhibits a transition from the 100 wt.% SS 316L region to a composite comprising 80 wt.% SS 316 and 20 wt.% Inconel 625, revealing predominant columnar dendrites oriented along the deposition direction. However, within the domain of 100 wt.% SS 316L, the microstructural features encompass austenite and skeletal-ferrite phases, aligning with prior research [22]. A notable alteration in micro-morphology is discernible at the bi-metallic interface between the 100 wt.% SS 316L and 80 wt.% SS 316L regions, as shown in Figure 2b. In the 80 wt.% SS 316L zone, the skeletal-ferrite phases are absent, and the microstructure transitions to the austenite solidification, attributed to the inclusion of Inconel 625, which provided abundant Ni elements and hindered the formation of δ-ferrite [23]. Within the 80 wt.% SS 316L/20 wt.% Inconel 625 region (Figure 2c), micro-cracks are present along the grain boundaries of differently oriented grains, primarily attributed to the accumulation of thermal and residual stresses at the grain boundaries [5]. 

Figure 2d,e illustrate the microstructures within the 60 wt.% SS 316L and 40 wt.% SS 316L regions, with the presence of Nb- and Mo-rich precipitates at the primary dendrite branch. As the sedimentation height increased, there was an observable upward trend in the spacing between the primary dendrite branches of columnar dendrites, indicating a coarsening of grains. This phenomenon arises from the tunability of the solidification morphology by the temperature gradient (G) with respect to the grain growth rate (R) ratio. Therefore, the G/R ratio serves as a crucial parameter in defining grain size, where a lower G/R value results in a columnar grain structure [24]. In the wire arc additive manufacturing (WAAM) process, heat dissipation primarily occurs through conduction to the substrate, with a minor contribution from thermal radiation into the surrounding atmosphere. As the deposition height increases, the distance between the molten pool of the specimen and the substrate increases, reducing the heat dissipation efficiency and cooling rate; this reduction promotes grain growth [25].

Figure 3 illustrates the microstructural transitions across the transitional zones of specimen #2. Within the 100 wt.% SS 316L region, the microstructure is composed of austenite and skeletal-ferrite phases. A notable shift in micro-morphology occurred when comparing the 100 wt.% SS 316L region to the 70 wt.% SS 316L zone. While the ferritic phase precipitated within the 100 wt.% SS 316L region dissolved at the bi-metallic interface between the 100 wt.% SS 316L and 70 wt.% SS 316L regions, the grains continued to grow along the original deposition direction. This indicates that changes in composition can affect the phase structure, but do not affect grain growth. Grain size and orientation are predominantly determined by the cooling rate [26]. Within the 70 wt.% SS 316L region, micro-cracks are absent, but granular Laves phase precipitates can be observed. This was mainly attributed to the increased content of Inconel 625, which led to the higher amounts of Nb and Mo elements as the predominant constituents of the Laves phase.

Figure 4 shows the microstructural variations across different compositional zones in specimens #3, predominantly characterized by columnar dendrites aligned with the deposition direction. Notably, the micro-morphology differs between the 100 wt.% SS 316L and 70 wt.% SS 316L regions, with the predominance of the austenite phase. However, at the bi-metallic interface, the absence of the ferritic phase allows for the precipitation of Laves phases [27]. Comparing Figure 2c and Figure 3c, it is evident that the amount and volume fraction of secondary phases increase within the first transitional gradient area, altering the precipitate morphology from lath-like to blocky with the increase in Inconel 625 content, primarily due to the abundant Nb and Mo elements as essential constituents of the Laves phases. Although the advent of these precipitates attenuated the solid solution strengthening effect, the increase in the second phase content enhanced the precipitation strengthening effect in the transition region to a certain extent [28].

In order to further study the microstructural evolution of FGMs, EBSD analysis was employed in this study. The grain structure of FGMs typically exhibits dendritic or cellular crystal formations, with predominant growth along the building direction. Figure 5 illustrates the Y0 inverse pole figure (IPF) maps of cross sections at the bi-metallic interface, focusing on the 100 wt.% SS 316L region and the first transition region for specimens 1, 2 and 3. A strong <001> texture aligned with the building direction is observed across all specimens due to the predominant heat dissipation occurring through conduction to the substrate [20]. In addition, the EBSD results reveal a consistent epitaxial grain growth at the bi-metallic interface across varying compositional regions, owing to the shared crystal structure (FCC) and chemical composition of SS 316L and Inconel 625 alloy [29]. Within the 100 wt.% SS 316L region, the granular grains can be observed at the grain boundaries, representing ferrite phases. Notably, with the addition of Inconel 625, the ferrite phase dissolves. Figure 5d presents the IPF figure of cracks in the 80 wt.% SS 316L/20 wt.% Inconel 625 region of specimen #1, revealing crack initiation at grain boundaries with disparate orientations [5]. Previous research has indicated that the addition of Inconel 625 leads to ferrite dissolution; the solubility of C element in the austenite phase decreases and the precipitation of MC phase increases, consequently predisposing the material to crack formation [21].

### 3.2. Composition Distribution 

Figure 6 depicts the elemental distribution transitioning from the 100 wt.% SS 316L region to the first transition gradient zone. Figure 6a depicts the elemental transition for specimen #1, shifting from 100 wt.% SS 316L to 80 wt.% SS 316L. Despite the notable divergence in microstructural morphology characterized by skeletal-ferrite phases exclusively in the 100 wt.% SS 316L region, the elemental distribution remained largely consistent. A small decrease in Fe element content and a slight increase in Ni were observed from 100 wt.% SS 316L to 80 wt.% SS 316L, with no significant alterations in Cr, Nb and Mo. In Figure 6b, the elemental composition for specimen #2 is pronounced upon the transition from 100 wt.% SS 316L to 70 wt.% SS 316L. Compared with specimen #1, considerable variation in element distribution across different compositional regions is evident. With the addition of Inconel 625, a substantial decrease in Fe element content is observed, while those of Ni, Nb and Mo increase. Moreover, the 70 wt.% SS 316L/30 wt.% Inconel 625 region exhibits heterogeneous distributions of Nb and Mo, attributed to the precipitation of the second phase observed in Figure 2, Figure 3 and Figure 4. Expanding the composition gradient from 100 wt.% SS 316L to 50 wt.% SS 316L/50 wt.% Inconel 625, as seen in specimen #3 (Figure 6c), further amplifies the changes in element distribution within the transition region compared to specimen #2. The line scans in Figure 6a1,b1,c1 delineate the elemental composition within the transition zones, revealing a progressive increase in the variation of compositional elements with the gradient. In addition, the line scans exhibit sharp peaks in the distribution of Nb and Mo elements within the gradient region, with a higher content of Inconel 625 corresponding to an increased presence of protrusions, indicating the abundant second phase inclusions.

### 3.3. Mechanical Properties

#### 3.3.1. Microhardness

Figure 7 depicts the microhardness distributions along the deposition direction for specimens #1, #2 and #3 at different compositional gradients. According to Figure 7a, despite specimen #1 exhibiting a transition from 100 wt.% SS 316L to 100 wt.% Inconel 625 with a uniform gradient of 20 wt.%, the microhardness first decreased and then increased along the deposition direction. Particularly, within the 80 wt.% SS 316L/20 wt.% Inconel 625 region, the microhardness reached its minimum value (177.7 ± 4.9 HV). Two factors may have contributed to this result. Firstly, the addition of a small amount of Inconel 625 can impede the formation of ferrite and the transformation of primary ferrite into austenite, which is essentially soft [30]. Secondly, the segregation of reinforcing elements (e.g., Nb and Mo) promotes the formation of a second phase, while the reductions in Mo and Nb in the austenite phase lead to a weakened solid solution strengthening effect, aligning with the findings of Su et al. [11]. The microhardness distribution within specimen #2 is shown in Figure 7b, where the 70 wt.% SS 316L region exhibits an increase in microhardness compared to the 80 wt.% SS 316L zone. The microhardness of the 70 wt.% SS 316L composition area is basically the same as that of the 100 wt.% SS 316L region, but the microstructure is significantly different. This variation is attributed to the increased Nb and Mo element contents in the Inconel 625 region, enhancing the solid solution strengthening and precipitation strengthening effects, leading to a rise in microhardness [31]. The microhardness distribution along the deposition direction for specimen #3 is illustrated in Figure 7c. It can be observed that the microhardness gradually increases along the deposition direction. Particularly noteworthy is that the microhardness within the 50 wt.% SS 316L region exceeds that across the 100 wt.% SS 316L region. With the incremental increase in Inconel 625 content in the subsequent transition region, the microhardness also increases. The microhardness distributions in specimens #1, #2 and #3 from the 100 wt.% SS 316L to 100 wt.% Inconel 625 zones reveal the overall improvement in the microhardness of FGMs through excluding non-beneficial compositional regions.

#### 3.3.2. Tensile Properties

Figure 8a depicts the tensile stress–strain curves at different compositional gradients, which exhibit severe plastic deformation of the tensile specimens prior to fracture, indicating a ductile fracture mechanism. The corresponding ultimate tensile strength (UTS), yield strength (YS) and elongation at break for these specimens are detailed in Figure 8b. Specifically, the values obtained for specimen #1 are 567.5 ± 7.6 MPa (UTS), 376.5 ± 20.8 MPa (YS) and 37.7 ± 1.6% (elongation at break). Those for specimen #2 are 590.6 ± 8.7 MPa (UTS), 392.4 ± 4.7 MPa (YS) and 39.6 ± 1.8% (elongation at break). Finally, specimen #3 exhibits the highest tensile properties, namely 631.8 ± 15.4 MPa (UTS), 423.1 ± 12.0 MPa (YS) and 41.6 ± 4.0% (elongation at break). These findings reveal that specimen #1 had the weakest tensile performance. Although the tensile strength and elongation at break of specimen #2 are slightly higher than those of specimen #1, the yield strength is still lower than that of SS 316L fabricated using the same process by Wang et al. [32]. Specimen #3 demonstrates the best tensile properties, surpassing those of 100 wt.% SS 316L. The diminished tensile properties of specimen #1 can be attributed to several factors: (1) the emergence of cracks within the 80 wt.% SS316L/20 wt.% Inconel 625 region; (2) the formation of brittle second-phase precipitates potentially compromising grain interface integrity, resulting in poor overall tensile performance of specimen #1; and (3) the loss of the ferrite phase [29]. As the content of Inconel 625 increased to 30 wt.%, the disappearance of cracks in the transition region and the increase in Nb and Mo element amounts enhanced the solid solution strengthening effect, leading to better tensile properties for specimen #2 compared to specimen #1. Nevertheless, the absence of a ferrite phase in the 30 wt.% Inconel 625 region resulted in a decline in mechanical properties. Additionally, the low Mo and Nb content and segregation leading to the formation of Laves phases limited the solid solution strengthening effect, resulting in inferior tensile properties compared to the 100 wt.% SS 316L region. For specimen #3, the first compositional transition gradient shows a further increase in Mo and Nb contents in the austenite phase. The effect of Mo and Nb elements on solid solution strengthening is the main reason for the increase in tensile strength. Consequently, both the tensile strength and elongation at break improved, exceeding those of the 100 wt.% SS 316L region. This indicates that the tensile performance of FGMs can be effectively enhanced through the optimization strategy by removing compositional regions that negatively affect the overall tensile properties.

The fracture morphologies of specimens #1, #2 and #3 are shown in Figure 9, featuring ductile dimples of assorted dimensions, which are typical of a ductile fracture through micro-void coalescence. Figure 9a illustrates the fracture of specimen #1, revealing the presence of micro-cracks that significantly impact tensile strength. In addition, a small amount of the second phase is detected within the depths of the dimple. In Figure 9b, the fracture morphology of specimen #2, exhibits a dendritic structure. Compared to specimen #1, the ductile dimples of specimen #2 appear finer and denser, with no observed cracks, which endows specimen #2 with better tensile properties. Figure 9c displays the fracture morphology of specimen #3, characterized by a prominent dendritic texture, devoid of defects (such as cracks, pores and so on). The dimples, indicative of toughness, become finer, denser and deeper, thereby enhancing the tensile performance. 

## 4. Conclusions

In this study, SS 316L/Inconel 625 FGMs were fabricated using the D-WAAM technique. The microstructures and mechanical properties of the materials within the compositional transition zones were meticulously evaluated to identify and exclude the regions with poor performance. The feasibility of this optimization strategy was explored both experimentally and theoretically to enhance the overall performance of FGMs. Based on the findings, the main conclusions can be drawn as follows.

(1) SS 316L/Inconel 625 FGMs with different composition grades were successfully prepared at different wire feed speeds. Despite the varying proportions of Inconel 625 in the initial layer transition gradient, all of the specimens demonstrated good metallurgical bonding at the bi-metallic interface.

(2) The microstructures of specimens #1, #2 and #3 consisted primarily of columnar dendrites and limited equiaxed grains. In the 100 wt.% SS 316L region, the microstructure presented a mixture of austenite and ferrite phases. In other regions, the ferrite phase disappeared, and the microstructure was predominantly composed of austenite with some second phase precipitates. The amount of precipitated phase increased with the increase in Inconel 625. Additionally, defects such as microscopic cracks were observed when Inconel 625 (80 wt.% SS 316L) was added in small amounts, and these cracks disappeared when this area was removed.

(3) With the gradual increase in Inconel 625 content, corresponding changes in microhardness and tensile properties were observed, although these changes did not linearly increase. Within the 80 wt.% SS 316L/20 wt.% Inconel 625 region, the mechanical properties decreased, especially microhardness in the gradient zone, resulting in a drop in tensile strength.

(4) The application of the optimization strategy to exclude areas with poor mechanical properties led to an enhancement in the overall mechanical performance of the FGMs. Initially, the microhardness distribution along the transition direction first decreased and then gradually increased in specimen #1. Furthermore, the tensile assessments reveal that specimen #3 exhibits superior ultimate tensile strength, yield strength and elongation at break (631.8 ± 15.4 MPa, 423.1 ± 12.0 MPa and 41.6 ± 4.0%, respectively).

In summary, the optimization strategy of omitting compositionally suboptimal zones provides valuable insights for the design and fabrication of FGMs using the D-WAAM process. Overall, the proposed method demonstrates the potential to extend the application of SS 316L/Inconel 625 FGMs. Future research will focus on further investigating the impact of WAAM process parameters on the overall performance of FGMs and optimizing their performance.

## Figures and Tables

**Figure 1 materials-17-02910-f001:**
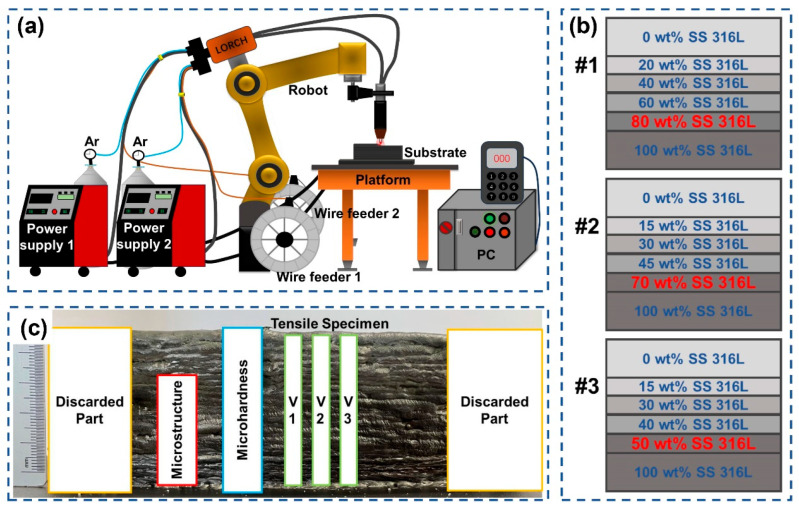
(**a**) Schematic diagram of a D-WAAM system, (**b**) the schematic of composition gradient design and (**c**) the sampling location of analyzed specimens.

**Figure 2 materials-17-02910-f002:**
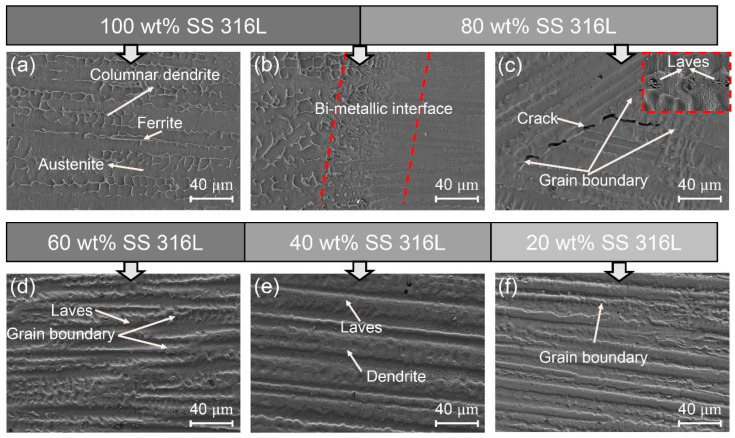
The microstructures of specimen #1: (**a**) 100 wt.% SS 316L; (**b**) bi-metallic interface of 100 wt.% SS 316L and 80 wt.% SS 316L; (**c**) 80 wt.% SS 316L; (**d**) 60 wt.% SS 316L; (**e**) 40 wt.% SS 316L; and (**f**) 20 wt.% SS 316L.

**Figure 3 materials-17-02910-f003:**
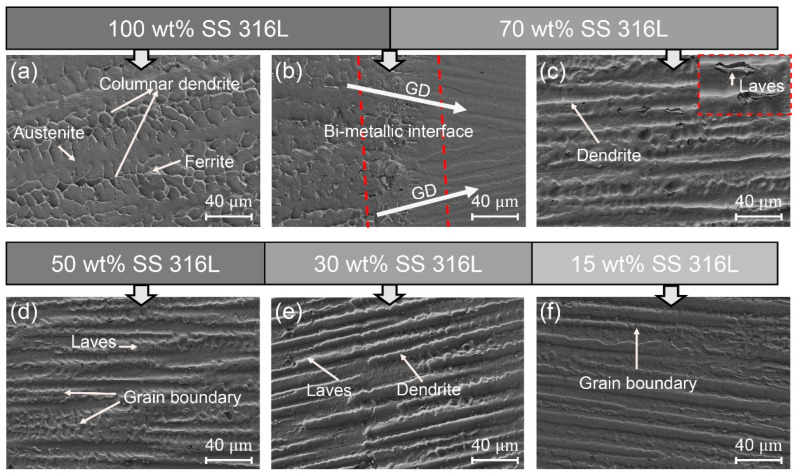
The SEM morphology of specimen #2: (**a**) 100 wt.% SS 316L; (**b**) bi-metallic interface of 100 wt.% SS 316L and 70 wt.% SS 316L; (**c**) 70 wt.% SS 316L; (**d**) 45 wt.% SS 316L; (**e**) 30 wt.% SS 316L; and (**f**) 15 wt.% SS 316L.

**Figure 4 materials-17-02910-f004:**
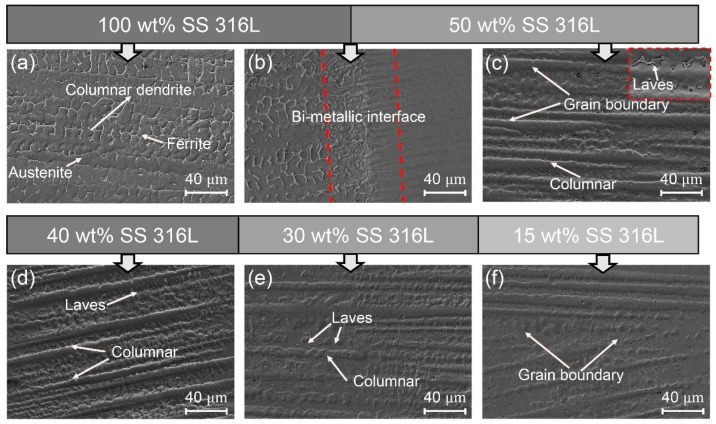
The SEM morphology of specimen #3: (**a**) 100 wt.% SS 316L; (**b**) bi-metallic interface of 100 wt.% SS 316L and 50 wt.% SS316L; (**c**) 50 wt.% SS 316L; (**d**) 40 wt.% SS 316L; (**e**) 30 wt.% SS 316L; and (**f**) 15 wt.% SS 316L.

**Figure 5 materials-17-02910-f005:**
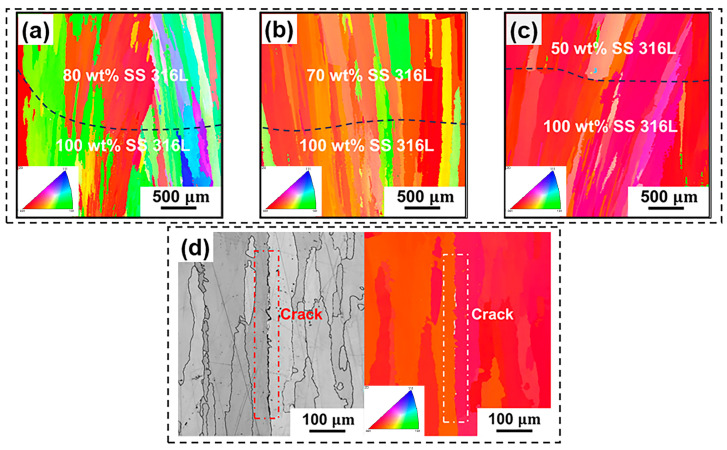
EBSD examination results of the bi-metallic interface in the 100 wt.% SS 316L region and the first transition region: (**a**) specimen #1, (**b**) specimen #2, (**c**) specimen #3 and (**d**) the 80 wt.% SS 316L region of specimen #1.

**Figure 6 materials-17-02910-f006:**
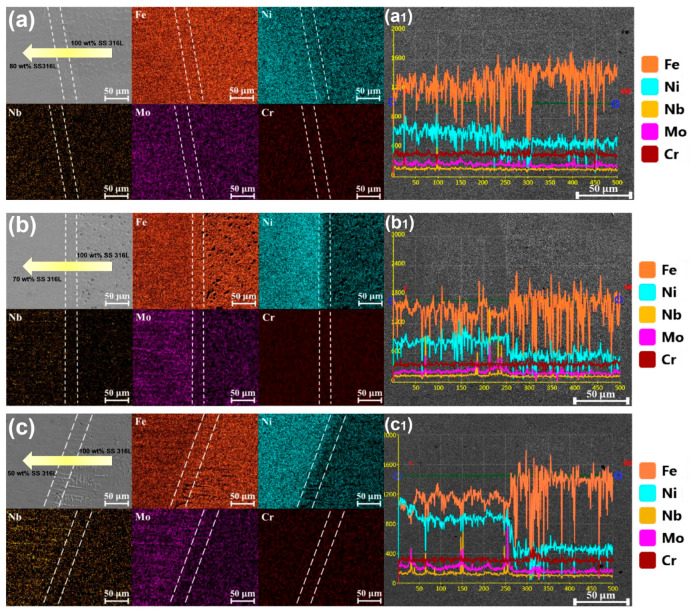
Elemental distributions of surface and line distribution maps at the bi-metallic interface from the 100 wt.% SS 316L zone to the first transition gradient region under different compositional gradients: (**a**,**a1**) specimen #1, (**b**,**b1**) specimen #2 and (**c**,**c1**) specimen #3.

**Figure 7 materials-17-02910-f007:**
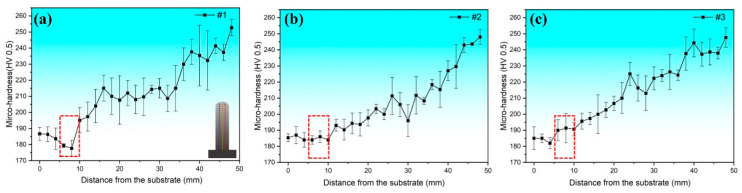
Microhardness distributions along the deposition direction under different compositional gradients: (**a**) specimen #1, (**b**) specimen #2 and (**c**) specimen #3.

**Figure 8 materials-17-02910-f008:**
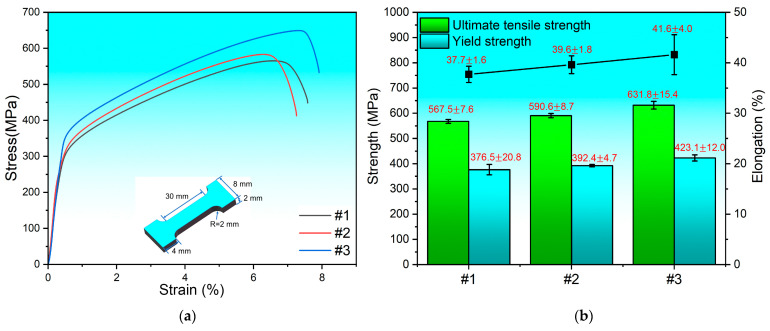
(**a**) Stress–strain curves and (**b**) tensile strengths of specimens under different compositional gradients.

**Figure 9 materials-17-02910-f009:**
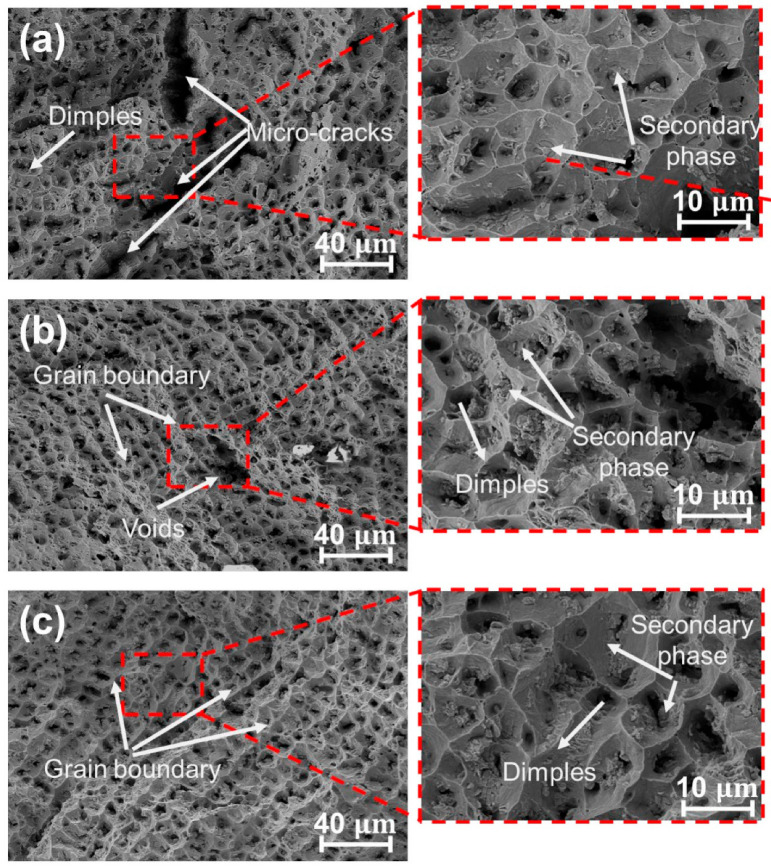
Tensile fracture morphologies of specimens (**a**) #1, (**b**) #2 and (**c**) #3.

**Table 1 materials-17-02910-t001:** Chemical compositions of substrate and welding wires (wt.%) [20].

Element	Cr	Ni	Mn	Si	C	Mo	Ti	Nb	Fe	Co	Al
SS 304L	18.5	8.0	1.7	0.22	0.03	-	-	-	Bal.	-	-
SS 316L	17	13.5	1.0	0.5	0.02	2.5	-	-	Bal.	0.3	0.09
Inconel 625	21	Bal.	0.36	0.36	<0.01	8.7	0.07	3.4	<0.1	-	-

**Table 2 materials-17-02910-t002:** Arc deposition experiment parameters of D-WAAM process.

Specimen	Layers	Wire Feed Speed (m/mm)		Travel Speed (cm/min)	Composition (SS 316L, wt%)	Current I/A	Welding Torch Height (mm)	Cooling Time (s)
		SS 316L	Inconel 625					
#1	1–10	6	0	60	1.00	110	15	30
	10–15	5	1.2	62	0.80			
	16–20	4	2.5	65	0.60			
	21–25	2.5	3.5	60	0.40			
	26–30	1.3	5	63	0.20			
	31–40	0	6	60	0.00			
#2	1–10	6	0	60	1.00	110	15	30
	10–15	4.5	1.8	63	0.70			
	16–20	3.2	3	62	0.50			
	21–25	2	4.4	64	0.30			
	26–30	1	5.2	62	0.15			
	31–40	0	6	60	0.00			
#3	1–10	6	0	60	1.00	110	15	30
	10–15	3.2	3	62	0.50			
	16–20	2.6	3.7	63	0.40			
	21–25	2	4.5	65	0.30			
	26–30	1	5.2	62	0.15			
	31–40	0	2.5	60	0.00			

## Data Availability

The raw data supporting the conclusions of this article will be made available by the authors on request.

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
