# Peer review of "Study on Optimization Strategy for the Composition Transition Gradient in SS 316L/Inconel 625 Functionally Graded Materials"

_materials, 2024, doi:10.3390/ma17122910_

Round 1
Reviewer 1 Report
Comments and Suggestions for Authors
I pretty much enjoyed the article and the systematic work the authors did.
Because their findings are quite interesting I would like to ask the authors to provide any possible application their work might have in the Defence industry, as the results they present here, I am sure they are very relevant.
Other than the comment above, I do not have any more comments.
Reviewer 2 Report
Comments and Suggestions for Authors
The subject of the manuscript is the fabrication of SS 316L/Inconel 625 functionally graded materials (FGMs) using Dual-Wire Arc Additive Manufacturing (D-WAAM) technology and the investigation of an optimisation strategy to improve the mechanical properties of these FGMs by excluding compositional regions prone to defects and poor performance.
1. The introduction provides a good background on functionally graded materials (FGMs) and their applications and the motivation for combining stainless steel and nickel-based superalloys. However, the discussion on the specific challenges associated with the Wire Arc Additive Manufacturing (WAAM) process for FGM fabrication could be expanded further.
2. The authors have not adequately justified the significance and novelty of their work. The introduction does not clearly articulate the research gap that this study aims to address. The introduction could benefit from a more detailed literature review of the existing research on optimising composition transition gradients in FGMs, particularly those produced via WAAM. This would help establish the novelty and significance of the current work more effectively.
3. The experimental setup and the details of the Dual-Wire Arc Additive Manufacturing (D-WAAM) process are well described. However, the authors should provide more information on the specific parameters used, such as the welding current, voltage, travel speed, and shielding gas flow rate, as these can significantly influence the microstructure and mechanical properties of the fabricated FGMs.
4. The authors should clarify the rationale behind selecting the specific transition gradient compositions (20 wt.%, 30 wt.%, and 50 wt.% Inconel 625) and explain how these were determined.
5. The discussion on the role of the Laves phase and the dissolution of the ferrite phase is superficial and lacks quantitative support. The authors have not convincingly explained the observed decrease in mechanical properties within the 20 wt.% Inconel 625 transition region.
6. The presented data do not adequately support the authors' claims regarding the effectiveness of the optimisation strategy. The discussion on the limitations and potential drawbacks of the approach is severely lacking. The authors should discuss the possible limitations of the optimisation strategy, such as the possibility of introducing other defects or microstructural heterogeneities due to the increased remelting and thermal history associated with the WAAM process.
Comments on the Quality of English LanguageThe manuscript has some English grammar, syntax, and typo issues. The manuscript requires thorough proofreading to address these language issues.
Reviewer 3 Report
Comments and Suggestions for Authors
The aim of the paper submitted for review is to produce FGM made of 316L stainless steel/Inconel 625 with a varied transitional composition using Dual-Wire Arc Additive Manufacturing (D-WAAM). The study includes a comprehensive evaluation of microstructural features, element distribution, and mechanical properties of FGMs under different transition gradient scenarios. The results obtained by the authors revealed grain boundary cracking and the lowest microhardness in the transition gradient zone containing 20 wt.% Inconel 625. According to the tensile tests, this was unfavorable for the overall mechanical properties of FGMs, exhibiting the lowest ultimate tensile strength (UTS), yield strength (YS), and elongation at break. After increasing the Inconel 625 content in the first layer of the transition zone, the authors observed that cracks disappeared, and microhardness increased, resulting in better tensile strength. The best mechanical properties were obtained with a 50 wt.% Inconel 625 content.
The authors of the reviewed paper concluded that the composition gradient optimization strategy, which eliminates component regions with poor mechanical properties and microdefects, proved effective, ensuring excellent mechanical properties of the final material.
The paper contains an excellent introduction – congratulations to the authors on this point.
Abstract – it should encourage reading, not indicate theoretical elements, and should not contain any results. The abstract should briefly and concisely indicate what the paper will be about. Currently, I consider the abstract too long – please shorten it.
The paper must include a nomenclature – a complete list of symbols, abbreviations, and designations. It can be placed even at the end of the paper. This is necessary – without it, I will not agree to the publication of the article.
Units on figures or tables should be written in square brackets, not round brackets. When referring to the test material, the term "specimen" should be used – not "sample", in accordance with ASTM, BS, or ISO standards. Please change this in the paper.
The scientific article should not be labeled as "work". It should definitely not be written this way. I recommend using terms like "paper, manuscript, scientific article, study, etc." Please correct this in the paper.
Equation No. (1) – please use a different notation for the multiplication sign – instead of the symbol "x", use the symbol "·".
Table 1 – please indicate the source – citation – where the authors obtained these data for the table, or whether they determined them themselves.
Please include specific technical drawings of the base material and specimens used in the tests.
I have no comments on the microstructure research – congratulations to the authors on the descriptions and results obtained.
However, regarding the mechanical properties – and essentially also the microstructure, I have some doubts. In my opinion, for each type of specimen - #1, #2, #3, the authors should prepare at least three specimens, or even at least five specimens. It should be shown for each specimen, how microhardness measurement is performed – how microhardness changes with distance – how it is measured and where it is measured. Several measurements should be shown and comparisons made for similarly made specimens to confirm the results. This would allow assessing the repeatability of the method and the accuracy of the specimen preparation. This will enable the assessment of dispersion, minimum, maximum, average values, median, and standard deviation. This is missing in the paper. In Figure 7, I miss an attempt to determine the relationship, how microhardness changes with distance from the substrate. It begs to draw a trend curve and provide the coefficient of determination. Perhaps it could then be related to some parameters determining the specimen preparation process. It almost certainly would.
Regarding the uniaxial tensile test – I have many comments. Figure 8 – strain is dimensionless – it has no unit, i.e. "mm/mm". Strain is expressed as "mm/mm" – dimensionless, or as a percentage. Please correct this necessarily. In the case of the tensile test, it should be stated what signals were recorded during the tests, how the force and displacement were measured – from the crosshead or extensometer (if so – what was the base of the extensometer), what these signals were used for – to determine what, and which formulas were used for this purpose. This is missing in the paper. To make the results reliable, there must be at least three tests under each condition, or at least five. This allows for a full statistical analysis of the results – minimum, maximum, average, median, dispersion, standard deviation, number of specimens. This is missing in the paper. For the recorded signals, for each specimen preparation condition, comparative graphs should be made to assess the repeatability of the measurements. This is missing in the paper. In Figure 8b, there is clearly a lack of an attempt to evaluate the functional relationship of yield strength, tensile strength, or elongation depending on certain parameters conditioning the specimen preparation process. I would also add the evaluation of Young's modulus, breaking stresses, and parameters in the Ramberg-Osgood law, corresponding to the material strengthening description – this is missing in the paper.
Furthermore, there is no reference in the paper to a specimen made solely of SS 304L material. This would show how the procedure implemented by the authors affects the change in material properties. But this is missing in the paper. It should be supplemented.
The conclusions in the paper should be expanded. Directions for further research should be indicated, how to use the obtained results in solving practical engineering problems.
I suggest a major revision. Please implement all my corrections and submit the paper for re-review.
Comments on the Quality of English LanguageMinor editing of English language required.
Round 2
Reviewer 3 Report
Comments and Suggestions for Authors
The authors have addressed almost all of my previous comments in the resubmitted version of the paper.
As a result, the paper has become more valuable and readable, likely to capture the interest of readers.
Overall, the paper is engaging, and I have no further substantive comments to make.
Congratulations to the authors. I recommend the paper for publication.
Comments on the Quality of English LanguageMinor editing of English language required.